# Ultrasound Assessment and Self-Perception of Pelvic Floor Muscle Function in Women with Stress Urinary Incontinence in Different Positions

**DOI:** 10.3390/diagnostics14192230

**Published:** 2024-10-06

**Authors:** Noa Krasnopolsky, Noa Ben Ami, Gali Dar

**Affiliations:** 1Department of Physical Therapy, Faculty of Social Welfare & Health Sciences, University of Haifa, Haifa 3498838, Israel; noakr3@gmail.com; 2Department of Physical Therapy, School of Health Sciences, Ariel University, Ariel 4070000, Israel; noaba@ariel.ac.il

**Keywords:** ICIQ-SF, ultrasound, positions, stress urinary incontinence (SUI), pelvic floor contraction

## Abstract

**Objectives**: This study analyzed the effect of different positions on pelvic floor muscle (PFM) function in women with and without stress urinary incontinence (SUI). **Methods**: This study included women with (*n* = 17, research group) and without (*n* = 25, control group) SUI. Using abdominal ultrasound, PFM function (maximum contraction and endurance) was measured in four different positions: lying, sitting, standing and squatting. The level of difficulty perceived by the participants was recorded. **Results**: In both groups, the best contraction was observed in the standing position and the weakest in the lying position. Women with SUI showed a lower ability to perform PFM contraction. A significant difference was found between the groups in the sitting and standing positions, and it was smaller in the research group. In the research group, the contraction displacement during sitting was 2.68 (1.67) mm versus 4.51 (2.62) mm in the control. The displacement during standing was 6.92 (3.50) mm versus 9.18 (5.05) mm, respectively (*p* < 0.05). In the research group, 52.9% reported the sitting position as the most difficult compared with only 12% in the control group. **Conclusions**: Women with SUI have lower PFM function while standing or sitting, but not while lying, than those without SUI. Variations in PFM function across different positions exist. A new protocol for PFM examination should be written with the standing position included.

## 1. Introduction

Pelvic floor muscles are important for women’s health as they control bladder and bowel continence and evacuation [1]. Thus, pelvic floor disorders might result from improper contraction of pelvic floor muscles. Stress urinary incontinence (SUI) is the most common type of urinary incontinence and occurs when abdominal pressure increases and continence is no longer maintained [2]. Urinary incontinence can have significant implications on well-being, participation in social and sports activities, lower self-esteem, inactivity and urinary tract infection [2,3].

Traditionally, PFM function is assessed in the lying position. It was previously found that the position of the body affects the contraction of PFMs in healthy women. Contraction intensifies from lying down to sitting and standing [4,5,6]. Several studies have compared pelvic floor function in healthy women in different positions [5,7,8,9]. Other studies compared healthy women and women suffering from SUI in the lying position [10,11,12] and recently also women suffering from SUI [13,14]. Studies conducted in vertical and lying positions concluded that gravity’s influence on PFMs is greater in vertical positions compared to while lying [1,9]. The excursion of the PFMs, as measured by transabdominal ultrasound, was largest in the standing position. However, this may not reflect an anatomically higher final position of the PFMs with contraction; rather, it may reflect a greater amplitude of movement due to the lower starting position [1,9]. Although PFM function is not identical during all positions, most of the examinations and research are carried out during the lying position.

There are several methods available for the assessment of PFM function. One of the common methods is abdominal ultrasound, as it is non-invasive and simple to use. Abdominal ultrasound is a reliable method for assessing PFM function and its measurements were found to be correlated with other methods, such as transperineal ultrasound, digital palpation and perineometry [15,16,17]. Furthermore, the ultrasound can indicate in real time if the participant is performing a correct contraction, which is demonstrated in an upward movement of the bladder; no contraction (no movement of bladder base); or incorrect contraction, which is demonstrated in a downward movement of the ladder [18]. Clinically, the device provides good visual feedback to the examiner and the examinee [16]. A disadvantage is the lack of a reference point, meaning the degree of PFM movement is determined in relation to a mobile starting point [18].

As SUI occurs during different activities in different positions, it is essential to examine PFM function in different positions in women suffering from SUI. This provides insight into whether gravity facilitates PFM contraction or, on the contrary, makes it more difficult for these muscles to contract properly.

The main aim of this study was to compare PFM function in the lying, sitting, standing and squatting positions between healthy women and women suffering from SUI. In addition, we aimed to compare the subjective evaluation of each participant regarding the difficulty of contracting the PFM in each position between the groups and to examine the relationship between incontinence severity and the ability to contract PFM. We hypothesize that women with SUI will have lower PFM function compared with those without SUI in all examined positions. We also hypothesize that pelvic floor muscle function will be better when standing as opposed to lying down.

## 2. Materials and Methods

### 2.1. Study Design

This observational case–control study was approved by the University of Haifa’s Institutional Review Board (#119/20). All the participants signed an informed written consent form prior to participation in this study, and then, the examination protocol was explained to them. The rights of the participants were protected. This study was registered prospectively on a public clinical trial database. Clinical trial registration no: NCT04288648 (clinicalTrials.gov, accessed on 28 February 2020).

### 2.2. Participants

The sample included 17 participants with SUI (research group) and 25 participants without SUI (control group) aged 21–45. The inclusion criterion for the research group was having a score higher than 0 in sections 3–5 according to the International Consultation on Incontinence Questionnaire—Short Form (ICIQ-SF), and for the control group, it was having a score of 0 in the ICIQ-SF. Participants were excluded from this study if they suffered from other types of urinary incontinence, vaginal or urinary tract infection, a neurological condition, urological, gynecological or any abdominal surgery, pelvic organ prolapse, BMI (body mass index) ≥ 30, pregnancy, chronic cough, constipation, unbalanced diabetes, medication affecting urination or had received pelvic floor physical therapy recently or in the past.

### 2.3. Study Procedure

All participants completed a demographic questionnaire (age, height, weight, number of pregnancies and births, no. of vaginal/cesarian births and physical activity during the week) and the ICIQ-SF. The ICIQ-SF is a self-administered questionnaire that provides a numerical score indicating the prevalence, severity and impact of urinary incontinence on an individual’s quality of life and type of incontinence. The scale ranges from 0 to 21, with a higher score indicating a greater severity of urinary incontinence [19,20,21]. Participants were then divided into research and control groups according to the ICIQ-SF results. Subsequently, a women’s health physiotherapist (N.K.) with 10 years of experience conducted examinations on all participants. Participants were asked to fill their bladder according to a protocol of consuming 600–750 mL of water in the course of one hour, finishing half an hour before the examination without voiding to allow for optimal imaging of the bladder [22].

PFM function was assessed via transabdominal ultrasound (Mindray DP 6600 portable device, Shenzhen, China) with a 6 MHz 35 mm curved transducer in four positions: lying, sitting, standing and squatting. The transabdominal ultrasound was used for this study as it is a reliable and valid method to assess PFM function according to bladder displacement. Furthermore, it allowed us to measure PFM function in various positions, such as sitting, which was not possible with other methods such as transperineal ultrasound.

For the lying position, the participant was in a supine position with a pillow under her head and her knees slightly flexed. For the sitting position, the participant was asked to sit on the edge of the examination bed with her feet on the floor or a small stool, slightly leaning backward on her hands. The standing position was performed next to the examination table for confidence, yet the participant was asked not to lean on or hold the bed. The squatting position was performed like the standing position, but the participant was asked to slightly flex her knees from a standing position.

The same sequence of testing positions was used for all participants.

In all positions, the ultrasound transducer was placed in the transverse plane immediately supra-pubically over the lower abdomen at a 15–30° angle from the horizontal plane. The bladder base was marked with the online caliper tool of the device [23,24]. Participants were directed to perform a pelvic floor muscle contraction and the examiner assessed if a correct contraction (upward displacement of the bladder base) was performed, and then measured bladder base displacement (in mm) (Figure 1). In each position, two contractions were maximum voluntary contractions, and the third contraction examined endurance. The participant was instructed to hold the contraction as long as she could (measured in sec). If the contraction exceeded 30 s, she was instructed to stop.

When the examination was completed, the examiner marked the hardest and the easiest positions for each participant according to the lowest/highest bladder base displacement and the contraction duration. Moreover, every participant was requested to specify what was the easiest and the hardest position in her opinion, on a scale of 0–3, 0 being the easiest. The initial instructions given to the participants were to “squeeze and lift” the PFMs. If the participant did not contract the PFMs correctly, other instructions or the ultrasound image were used to provide visual feedback until they managed to contract the PFMs in all positions. A minute of rest was given between each contraction. If the participants were unable to elevate the pelvic floor in one of the positions, they scored 0 in that position, but if they could not elevate the PFMs in all positions, they were withdrawn from this study and referred to a specialist.

### 2.4. Data Analysis

The analysis of data was performed using a statistical software program (R Foundation for Statistical Computing version (4.0.5)). The degree of movement served as the dependent variable. The different positions and type of research group were chosen as independent variables. The normality of distribution was checked before examining the differences between the research groups using Spearman correlation coefficients. The data were not normally distributed; therefore, the continuous variables were analyzed using the Wilcoxon two-sample test, and the categorical variables were analyzed using the chi-square test. Univariate analysis using a two-sample Wilcoxon test or chi-square test was performed to find the association between demographics, clinical and outcome measures, in addition to the group. Univariate analysis, using a two-sample Wilcoxon test, Kruskal–Wallis test or Spearman correlation coefficients, was performed in order to find any association between ICIQ score and demographics and outcome measures. Fleiss’ (1971) category-wise Kappa was computed using the kappam.fleiss function form R’s irr package to determine the overall agreement between objective and subjective and each one of the category values. Sample size calculation was performed using G*power software (version 3.1.9.7) for the main outcome measure of PFM function, as observed by bladder base displacement (in mm). Following our previous study [22] and the initial results of the current study, we assumed that the continent and the incontinent groups would have a mean score of 5.5 (±3) and 3 (±2) mm, respectively. Given these results, the required total sample size, under the assumption of alpha = 5% and power of 80%, was 36 subjects (18 in each group).

## 3. Results

### 3.1. Characteristics of Participants

Forty-two women participated in this study, twenty-five in the control group and seventeen in the research group. Three participants were excluded from this study: one with BMI > 30; the second suffered from a gynecological condition and had attended pelvic floor physical therapy in the past; and the third could not perform a contraction in any position despite all attempts.

The lowest score in the ICIQ-SF questionnaire in the research group was 3 (slight leakage degree), the highest was 17 (severe leakage degree) and the average score was 8.06 (±4.44) (moderate leakage). No statistically significant relationship was found between the score in the ICIQ-SF questionnaire and demographic characteristics or types of childbirths.

No differences were observed between the groups in terms of age, height, weight, BMI and physical activity. A significant difference was found in the number of pregnancies and births between groups, which was higher in the research group. The demographic characteristics of the research groups are described in Table 1.

### 3.2. PFM Function in Different Positions

A significant difference in PFM function, as assessed by bladder displacement via the transabdominal ultrasound, was found between the groups when sitting and standing, and it was smaller in the research group. In the research group, the contraction displacement measurement during sitting was 2.68 (1.67) mm versus 4.51 (2.62) mm in the control group; while when standing, the contraction displacement was 6.92 (3.50) mm versus 9.18 (5.05) mm, respectively (*p* < 0.05). Likewise, a significant difference was found in endurance contraction between the research and control group in the sitting position, 8 sec versus 30, respectively (*p* < 0.05).

In both groups, the highest displacement found was while standing. Nonetheless, the maximal contraction in the control group was higher compared to the research group. No significant difference was found for all other measurements (Table 2). Since the results were not normally distributed, we used the median index.

### 3.3. The Correlation between Subjective Reports of Perceived Contraction Difficulty and the Objective Results of the Contractions

Following the ultrasound exam, participants were asked to identify the most challenging position for contracting the PFMs. A significant difference was found between research groups in relation to the subjective assessment. In the research group, 52.9% reported the sitting position was the most difficult, 35.3% the lying position and 11.8% the squatting position. On the other hand, in the control group, only 12% of the participants reported that the sitting position was the most difficult, 72% reported the lying position and 16% the squatting position. Neither group reported the standing position as the most difficult (Table 3).

When examining the objective assessment via the ultrasound measurements for the research group, 58.8% of the participants had the weakest contraction during sitting, followed by 35.3% during lying and 5.88% during squatting. In the control group, 12% of the participants had the weakest contraction during sitting and squatting and 76% during lying.

A correlation between the subjective report for the hardest position and the objective contraction results for all four positions was performed by the Fleiss’ Kappa test. In the lying and sitting positions, the correlation was medium–strong (0.56 and 0.6, respectively), and in the squatting position, the correlation was weak (0.32). This implies that for these positions, the hardest position as determined by lower contraction measured with the ultrasound was the same as the hardest position reported by the participants.

### 3.4. Severity of Incontinence and PFM Function

When evaluating the severity of incontinence and PFM function in the research group, a correlation was found between the score in the ICIQ-SF questionnaire and PFM contraction during the lying and squatting positions (Spearman 0.5–0.6, *p* < 0.5). This implies that the higher the score in the questionnaire, the lower the average contraction and endurance in these positions.

## 4. Discussion

This study examined pelvic floor muscle (PFM) function in women with and without SUI across various positions. The findings reveal distinct differences in PFM function between the groups, which were particularly evident in the sitting and standing positions. This suggests diminished PFM function among participants experiencing UTIs.

Other studies that explored pelvic floor muscle (PFM) contraction in various positions used different methods or populations. Dewaele et al. (2018) [7] and Chmielewska et al. (2015) [8] used electromyography (EMG) for PFM assessment and exclusively included women without urinary incontinence. The population examined in the studies conducted by Frawley et al. (2006) [9] was physical therapists, some specializing in women’s health. This might introduce a potential bias that could impact the findings. Frawley et al.’s studies did not document symptoms or other limitations of pelvic floor function in the recruited participants. Among parous women in Frawley et al.’s studies, some experienced minor urinary incontinence symptoms, with or without pelvic floor organ prolapse, and the women were not categorized into research and control groups. In addition, Frawley et al. [9,25] utilized a comprehensive approach, employing tools such as manometry, digital muscle testing and abdominal ultrasound. Gimenez et al. [14] examined women with SUI in the supine and lying positions using vaginal palpation, manometry and EMG. Their results showed that the pressure and EMG activity during contraction and PFM function were lower when standing, while the resting assessments were higher when standing. Mastwyk et al. [13] also examined women with SUI in the supine and lying positions using digital palpation and manometry. Their findings showed higher squeeze pressure during standing, while when using digital muscle testing, it was lower in the standing position compared with the lying position.

Using abdominal ultrasound, Frawley et al. (2006) found greater pelvic floor muscle displacement during the standing position compared to the lying and sitting positions, with lying showing slightly less displacement than sitting. These findings align with our study, also conducted using abdominal ultrasound. We observed the most significant displacement of the bladder base during the standing position. Dewaele et al. (2018) [7], using EMG, identified significant variations in resting PFM tension across different positions. Higher tension was noted during standing, which might explain the increased ability for a stronger contraction in this position, aligning with our study and Frawley et al.’s (2006) findings. In addition, the standing position, in contrast to the sitting position, was identified as the most provocative for incontinence among most women experiencing these symptoms.

These findings highlight the impact of the testing position on the extent of contraction, emphasizing the importance of considering and adding various test positions in regular PFM evaluation. Hence, performing an examination in the standing position may provide a more accurate reflection of pelvic floor function.

Some differences between studies’ results can be attributed to the use of different measuring methods. The abdominal ultrasound device assesses the upward or downward displacement of the bladder, using the bladder base as a marker for PFM activity. In contrast, digital muscle testing evaluates both lift and closing pressure of the PFMs, while manometry focuses solely on closing pressure. The incompatibility between displacement values and muscle strength measured digitally suggests that these tools assess different aspects of PFM function, making it challenging to compare results obtained with different tools.

In addition to PFM function assessment, participants subjectively ranked the difficulty of each position for PFM contraction immediately post-examination. In the SUI group, the majority found the sitting position the most challenging, while the control group reported lying down as the most difficult. Interestingly, none of the participants considered the standing position the most challenging. Kelly et al. (2007) [1] proposed a hypothesis linking gravity to the tension–length ratio of PFMs, suggesting that the force of gravity enhances this ratio as pelvic organs exert downward pressure. This increased pressure on stomach organs and the pelvic floor during standing may elevate tension, leading to improved PFM strength. An alternative explanation for participants not ranking the standing position as the most challenging lies in the greater amplitude experienced in this position compared to others. This heightened amplitude served as sensory feedback, particularly when participants faced difficulty contracting their PFMs in the lying or sitting positions. Some participants struggled to produce a contraction while lying or sitting, even with additional explanations, but found success in contracting the PFMs while standing. This was attributed to a heightened sensory awareness of the PFM and the contraction in the standing position. In contrast, in the lying position, as opposed to vertical postures, gravity is experienced as primarily affecting the posterior wall of the abdominal cavity rather than the pelvic floor, resulting in smaller muscle tension and amplitude. The sitting position demonstrated a smaller amplitude compared to the standing position, and the added challenge in squatting was attributed to maintaining the position.

### Research Limitations

Our research has a few methodological limitations. The research was subject to bias as the same researcher conducted examinations and documented data. Additionally, participants sought feedback from the monitor when struggling to contract the pelvic floor, potentially introducing bias. The sample size was relatively small. Future studies should be conducted on a larger sample sizes and with blind assessors.

## 5. Conclusions

Women with SUI have lower PFM function while standing or sitting, but not while lying down, in comparison to those without SUI. Variations in PFM function across different positions are highest in the standing position and lowest in the lying position.

Clinical examination of PFM function often relies on lying positions and may not fully capture pelvic floor conditions which are mostly found in the standing position. Therefore, the standing position examination should be included in the examination protocol.

Our findings, alongside those of other studies, could serve as a foundation for future research aiming to develop comprehensive guidelines and protocols for assessing PFM function in diverse positions. This, in turn, could lead to personalized treatment plans to assist women with incontinence in resuming daily activities and sports.

## Figures and Tables

**Figure 1 diagnostics-14-02230-f001:**
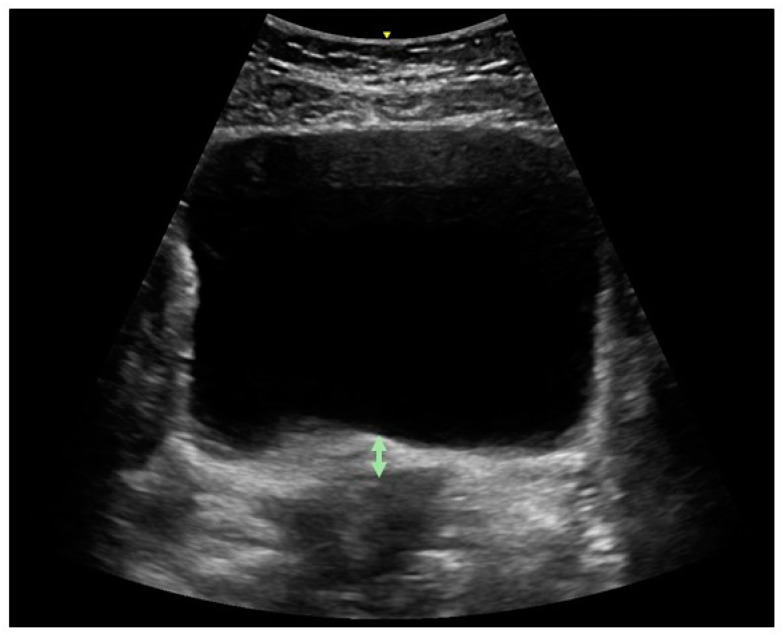
Ultrasound examination of pelvic floor muscle function as assessed by displacement of bladder base from rest to contraction. Upward movement occurs during correct contraction.

**Table 1 diagnostics-14-02230-t001:** Demographic characteristics.

Variables	Control Group (*n* = 25) Mean (±SD)	Research Group (*n* = 17) Mean (±SD)	*p*-Value
Age (years)	33.13 (±5.72)	34.41 (±7.09)	0.274
BMI	23.13 (±3.65)	24.12 (±3.86)	0.214
Height (Cm)	163.35 (±5.88)	161.82 (±4.69)	0.207
Weight (Kg)	61.79 (±10.73)	63.29 (±11.29)	0.434
Number of exercise sessions (per week)	2.13 (±1.62)	2.29 (±1.79)	0.663
Number of pregnancies	1.58 (±1.48)	2.24 (±1.52)	0.018 *
Number of births	1.43 (±1.24)	2.11 (±1.32)	0.006 *
Number of vaginal births	10	14	0.284
Number of cesarian births	7	4	0.284

(Some of the women had vaginal and cesarian births; therefore, they appear in both categories); BMI—body mass index (weight in kilograms divided by height in meters squared), *p* < 0.05 *.

**Table 2 diagnostics-14-02230-t002:** Pelvic floor contraction in different positions in both groups.

Position	Control Group (*n* = 25) M (Q1, Q3)	Research Group (*n* = 17) M (Q1, Q3)	*p*-Value
Maximal contraction lying (mm)	3.34 (2.31; 4.49)	2.50 (1.34; 3.57)	0.083
Endurance contraction lying (sec)	10 (0.00; 17.0)	6 (0.00; 30.0)	0.958
Maximal contraction sitting (mm)	4.51 (3.79; 6.84)	2.68 (1.69; 3.52)	0.001 *
Endurance contraction sitting (sec)	30 (18.0; 30.0)	8 (0.00; 17.0)	0.003 *
Maximal contraction standing (mm)	9.18 (6.74; 12.9)	6.92 (4.14; 11.1)	0.050 *
Endurance contraction standing (sec)	30 (20.0; 30.0)	30 (24.0; 30.0)	0.664
Maximal contraction squatting (mm)	7.14 (5.88; 10.8)	6.34 (5.36; 8.01)	0.081
Endurance contraction squatting (sec)	30 (25.0; 30.0)	25 (13.0; 30.0)	0.070

*p* < 0.05 *.

**Table 3 diagnostics-14-02230-t003:** Subjective reports and objective results regarding the hardest pelvic floor contraction position.

Position	Control Group (*n* = 25)				Research Group (*n* = 17)				Difference between Groups *p*-Value
	Lying *n* (%)	Sitting *n* (%)	Standing *n* (%)	Squatting *n* (%)	Lying *n* (%)	Sitting *n* (%)	Standing *n* (%)	Squatting *n* (%)	
**Subjective** **report**	18 (72%)	3 (12%)	-	4 (16%)	6 (35.3%)	9 (52.9%)	-	2 (11.8%)	0.013 *
**Objective** **results**	19 (76%)	3 (12%)	-	3 (12%)	6 (35.3%)	10 (58.8%)	-	1 (5.88%)	0.004 *

*p* < 0.05 *.

## Data Availability

The data that support the findings of this study are available from the corresponding author upon reasonable request.

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
