# Peer review of "Ultrasound Assessment and Self-Perception of Pelvic Floor Muscle Function in Women with Stress Urinary Incontinence in Different Positions"

_diagnostics, 2024, doi:10.3390/diagnostics14192230_

Round 1

Reviewer 1 Report

Comments and Suggestions for Authors

Line 51

To evaluate the excursion of the PFM, the transabdominal ultrasound technique was used. I wonder why not the transperineal one, which would have offered a better view but above all much more precise angles and evaluation lines, especially in conditions of a supine patient

The authors also confirm that one of the disadvantages is not having a fixed but mobile reference point with this technique, as instead you have in the transperineal technique (pubic bone)

For the rest, excellent research well conducted, with few methodological limitations.

Author Response

comment:

Line 51. To evaluate the excursion of the PFM, the transabdominal ultrasound technique was used. I wonder why not the transperineal one, which would have offered a better view but above all much more precise angles and evaluation lines, especially in conditions of a supine patient The authors also confirm that one of the disadvantages is not having a fixed but mobile reference point with this technique, as instead you have in the transperineal technique (pubic bone)

For the rest, excellent research well conducted, with few methodological limitations.

answer:

We thank the reviewer for the constructive approach and the positive feedback.

We used the trans-abdominal ultrasound for several reasons: first, it is a reliable method to assess pelvic floor muscle function in real-time during rest and contraction. Following this, it has several advantages mainly it is not an invasive assessment, and as such the participants feel more comfortable participating in the research. Second, this research examined several positions, such as sitting, standing, and squatting. If possible at all, it is much more difficult to perform the perineal ultrasound in these positions.

Following the comment, we have added this in the introduction (lines 51-54) and in the methods part  (107-110) for further clarification.

Reviewer 2 Report

Comments and Suggestions for Authors

I read with interest the proposed study on pelvic floor muscle (PFM) function in women, with and without SUI across various positions.
I must say that the title was promising, but while reading I was quite disappointed by the setup.
Unfortunately the good intentions clashed with the small overall number of subjects evaluated was only 42 subjects, 17 participants with SUI (research group) and 25 participants without SUI (control group).
Unfortunately the numbers are too small to draw any conclusions.
Was a calculation of the statistical power of the study done before starting it? If so, it should be included in the text.

Author Response

comment:

I read with interest the proposed study on pelvic floor muscle (PFM) function in women, with and without SUI across various positions.
I must say that the title was promising, but while reading I was quite disappointed by the setup.
Unfortunately the good intentions clashed with the small overall number of subjects evaluated was only 42 subjects, 17 participants with SUI (research group) and 25 participants without SUI (control group).
Unfortunately the numbers are too small to draw any conclusions.
Was a calculation of the statistical power of the study done before starting it? If so, it should be included in the text.

Answer: 

We thank the reviewer for the comment and appreciate the time spent reading the manuscript.

Sample size calculation was conducted before the research. We have added it in the analysis part. (lines 160-165). In addition, we added it to our limitation part (lines 299-300).

Furthermore, clinical studies are often conducted on relatively small sample sizes. Below are examples for published manuscripts on the topic of pelvic floor with relatively small sample sizes:

  1. Berube, M. E., & McLean, L. Differences in pelvic floor muscle morphology and function between female runners with and without running‐induced stress urinary incontinence. Neurourology and Urodynamics. 2023, 42(8), 1733-1744.‏ The sample included 19 and 20 participants in 2 research groups.
  2. Capson, A.C.; Nashed, J.; Mclean, L. The Role of Lumbopelvic Posture in Pelvic Floor Muscle Activation in Continent Women. Electromyogr. Kinesiol. 2011, 21, 166–177, doi:10.1016/j.jelekin.2010.07.017. – The sample included sixteen nulliparous, continent women.
  3. Bø, K.; Finckenhagen, H.B. Is There Any Difference in Measurement of Pelvic Floor Muscle Strength in Supine and Standing Position? Acta Obstet. Gynecol. Scand. 2003, 82, 1120–1124, doi:10.1046/j.1600-0412.2003.00240.x. – The ample included eighteen women, mean age 43.4 years (range 31-64 years), with symptoms of stress and mixed incontinence.

We hope that the reviewer will agree with the importance of this study and that the manuscript is now suitable for publication.